# Risk Factor and Replacement Therapy Analysis of Pre- and Postoperative Endocrine Deficiencies for Craniopharyngioma

**DOI:** 10.3390/cancers15020340

**Published:** 2023-01-04

**Authors:** Lidong Cheng, Hongtao Zhu, Jing Wang, Sisi Wu, Suojun Zhang, Junwen Wang, Kai Shu

**Affiliations:** Department of Neurosurgery, Tongji Hospital, Tongji Medical College, Huazhong University of Science and Technology, Wuhan 430030, China

**Keywords:** craniopharyngioma, hypopituitarism, surgery, pituitary stalk preservation, replacement therapy

## Abstract

**Simple Summary:**

Although craniopharyngiomas are pathologically defined as benign tumors, their lesions could involve the hypothalamic–pituitary axis and adjacent to several important anatomical structures. Treatment strategies, including surgery and radiation, have a higher risk of damaging these important structures, which leads to excess recurrence and disability rates. Endocrine deficiencies caused by hypothalamic–pituitary axis damage are the most common symptoms and postoperative complications of craniopharyngioma patients, which severely affect perioperative and long-term treatment effects. However, limited studies have focused on the endocrine deficiencies of craniopharyngioma patients. In this study, the influencing factors and replacement therapies for pre- and postoperative pituitary hormone deficiency were retrospectively analyzed in 126 craniopharyngioma patients, which provides evidence for surgical strategy selection and long-term management during craniopharyngioma treatments.

**Abstract:**

Background: Pituitary hormone deficiency (PHD) is one of the most common symptoms and postoperative complications of craniopharyngiomas (CPs). However, the risk factors for PHD in CPs are little known. The purpose of this study was to analyze the risk factors of pre- and postoperative PHD and to investigate replacement therapy for CP patients. Methods: A retrospective study of 126 patients diagnosed with CP was performed. Univariate analysis was performed using Pearson’s chi-squared test or Fisher’s exact test, and a multiple logistic binary regression model was used to identify the influencing factors of pre- and postoperative PHD in craniopharyngioma. Results: Children and patients with hypothalamic involvement were more likely to have preoperative PHD. Patients with suprasellar lesions had a high risk of postoperative PHD, and preoperative PHD was a risk factor for postoperative PHD. Conclusion: Children have a high incidence of preoperative PHD. Preoperative PHD can serve as an independent risk factor for postoperative PHD. Preoperative panhypopituitarism can serve as an indication of pituitary stalk sacrifice during surgery. The management of replacement therapy for long-term postoperative endocrine hormone deficiency in patients with craniopharyngioma should be enhanced.

## 1. Introduction

Craniopharyngiomas (CPs) originate from remnants of the craniopharyngeal duct epithelium and account for 1.2–4.6% of all brain tumor [1]. Patients with craniopharyngiomas have a bimodal age distribution, with peak incidence rates in children aged 5–14 years and in adults aged 50–74 years. Adamantinomatous craniopharyngioma (ACP) and papillary craniopharyngioma (PCP) are two histological subtypes of craniopharyngiomas. As CP is pathologically defined as a benign tumor (WHO I), it is curable when gross total resection is achieved. The 10-year survival rate of patients with CP ranges from 65% to 100%; however, lesions can involve any site of the hypothalamic–pituitary axis and are adjacent to several important anatomical structures, including the optic chiasma, hypothalamus, and main vessels [2,3]. Treatments, including surgery and radiation, have a higher risk of damaging these important structures, leading to excess recurrence and disability rates of CP.

Endocrine deficiencies are common symptoms and postoperative complications of CP that severely affect the perioperative and long-term treatment effects of patients [4]. Acute pituitary hormone deficiency, such as adrenal crisis and severe fluid and electrolyte imbalances, may lead to death. Long-term hormone deficiency can result in abnormal development in children and sexual dysfunction in adults, which significantly reduces long-term quality of life [5]. 

Previous research on CP has mainly focused on surgical techniques and life quality evaluation. Rare studies have paid attention to the risk factors and long-term management of endocrine hormone deficiencies in patients with CP. In this study, the clinical and follow-up data of 126 patients with CP were collected. We analyzed the factors influencing preoperative and postoperative pituitary hormone deficiency and investigated replacement treatment for hypopituitarism in these 126 patients, which can provide evidence for surgical strategies and long-term management in patients with CP.

## 2. Materials and Methods

### 2.1. Patients and Data

In total, 485 CP patients who underwent surgical treatment from 2010 to 2019 were retrospectively analyzed. All patients were diagnosed with CP based on a pathological examination. The inclusion criteria were as follows: (1) the pre- and postoperative clinical data, laboratory, and imaging examinations were complete; (2) the survival time more than 1 year; (3) without other endocrine diseases; (4) with physical condition can finish follow-up. Finally, 126 patients were included, and all patients were accurately followed up. Patient information, including age, sex, main symptoms, number of previous surgeries, intraoperative conditions, and pathological classification. Endocrine hormone and neuroimaging evaluations were performed in all patients pre- and post-surgery, re-examinations were conducted during follow-up, and the replacement therapies of each patient were recorded.

### 2.2. Endocrinological Evaluation

Endocrine hormones, including growth hormone (GH), prolactin (PRL), follicle-stimulating hormone (FSH), luteinizing hormone (LH), thyroid-stimulating hormone (TSH), adrenocorticotropic hormone (ACTH), cortisol, testosterone, estrogen, free T4, serum sodium, and urine-specific gravity, were routinely monitored during the perioperative period and reviewed at 1 month, 3 months, 6 months, and 1 year after surgery. The criteria for diagnosis in this study were as follows: (1) panhypopituitarism (CH, panhypopituitarism): three or more pituitary hormone deficiencies [6]. (2) Growth hormone deficiency (GHD): basal level of IGF1 lower than normal (434 ± 84 ng/mL), and peak GH level under stimulation < 10 ng/mL in children and <5 ng/mL in adults during the insulin hypoglycemia stimulation test [7]. For patients with clinical symptoms of GHD accompanied by a lower IGF1 level and three or more pituitary axis hormone deficiencies, diagnosis of GHD occurs directly [8]. (3) Hypothalamic–pituitary–thyroid axis deficiency (HPTD): free T4 lower than normal (0.7–1.48 ng/dL), accompanied by normal or decreased TSH level (0.6–4.5 mIU/L) [9]. (4) Hypothalamic–pituitary–adrenal axis deficiency (HPAD): basal serum cortisol < 3 µg/dL or peak level < 18 ng/mL in the ITT test [2]. (5) Gonadotrophin deficiency (GD): in males, serum testosterone level lower than normal (1.75–7.81 ng/mL) and low levels of FSH (1.27–19.26 mIU/mL) and LH (1.24–8.62 mIU/mL). In females, lower serum estradiol levels (20–47 pg/mL). In premenopausal women, amenorrhea and low levels of FSH and LH [2]. In postmenopausal women, without increased levels of FSH (16.74–113.59 mIU/mL) and LH (10.87–58.64 mIU/mL). (6) Diabetes insipidus (DI): With clinical manifestations of DI, laboratory test abnormalities include serum sodium (>145 mmol/L), urine specific gravity (<1.005), and urine amount (>250–300 mL/h for 2–3 h) [10]. Permanent DI was defined as persistent DI symptoms for more than 1 year, requiring medication to control the amount of urine.

### 2.3. Neuroradiological Evaluation

All patients underwent preoperative brain CT and MRI examinations. The location, size, texture, and involvement of the hypothalamus in the lesions were analyzed. Brain MRI was reexamined within 24 h and at 3 months, 6 months, and 1 year post-operation. Tumor size was calculated according to the maximum diameter based on MRI, and the patients were divided into ≤3 cm and >3 cm groups. According to the location of the lesion, we grouped the patients into intrasellar, suprasellar-extraventricular, and suprasellar-intraventricular types. Based on the tumor texture, patients were classified into three groups: solid, cystic-solid, and cystic. The relationship between the tumor and the hypothalamus of each patient was also classified into three grades according to the Puget grade: Grade 0, no hypothalamic involvement; Grade 1, the tumor abutting or displacing the hypothalamus; Grade 2, hypothalamic involvement (the hypothalamus is no longer identifiable) [11].

### 2.4. Treatment

Patients with preoperative thyroid hormone and cortisol deficiencies were routinely supplemented. In patients with acute hydrocephalus, external ventricular drainage should be performed first, if necessary. The surgical strategy depends on a preoperative imaging examination. There are two mainstream approaches: transsphenoidal approaches (TSA) and transcranial approaches (TCA). The intrasellar type is mainly resected via transsphenoidal endoscopy or microscopy. The pituitary stalk preservation standard means that the pituitary stalk can be identified and is intact during surgery. Complete resection was defined as the complete removal of the lesion under a microscope with no residual tumor or calcification on reimaging [12].

### 2.5. Follow-Up

The patients were followed up at 1, 3, and 6 months and every year postoperatively. Endocrine hormone and imaging examinations were performed at each follow-up visit. Pituitary hormone status, replacement therapy of patients, and tumor recurrence, if found, were recorded. Recurrence was defined as the growth of the residual tumor or a new lesion on follow-up MR imaging [3]. The follow-up time ranged from 12 to 111 months, with a mean of 54.1 ± 33 months.

### 2.6. Statistical Methods

Data were processed using IBM SPSS22.0 software (IBM Corp, Armonk, NY, USA). Mean values between groups were analyzed using the Student’s *t*-test. Pearson’s Chi-Squared (χ^2^) test or Fisher’s exact test was used for univariate analysis to describe the frequency and ratio of categorical variables. Variables with univariate analysis results *p* < 0.1 were included in the multivariate logistic regression model. A multivariate logistic binary regression model was used to determine independent risk factors that affect pituitary hormone deficiency in patients with CP pre- and post-surgery. A two-sided level of *p* < 0.05 was considered statistically significant.

## 3. Results

### 3.1. Patient Characteristics and Symptoms

The basic information and clinical characteristics of the 126 patients are shown in Table 1. In this study, 47 patients were female (37.3%). Patients’ ages ranged from 3 to 74 years, with an average of 35.4 ± 19.3 years, including 29 children (<18 years old, 23%) and 97 adults (≥18 years old, 77%). Approximately half of the patients showed symptoms related to intracranial hypertension (50.8%), such as headaches, nausea, and vomiting. Vision loss and visual field defects were observed in 59 (46.9%) patients. Twenty patients (15.9%) showed symptoms of hormone deficiency, including abnormal development, sexual dysfunction, amenorrhea, and diabetes insipidus. Twelve patients (9.6%) exhibited abnormal hypothalamic function, including abnormal sleep, weight gain, and personality changes. Another nine patients were incidentally discovered by regular check-ups, three patients for epilepsy, two patients for high fever, and one patient for a sudden loss of consciousness. Seventeen patients had a history of CP surgery, while the other 109 cases (86.5%) underwent surgery for the first time.

### 3.2. Neuroradiological Features

Tumors of 16 patients (12.7%) were intrasellar, and those of 110 patients (87.3%) were suprasellar, of which 46/110 (36.5%) had third or lateral ventricle involvement. The maximum diameter of the lesions ranged from 1.2 to 8.6 cm, 36.5% of them were less than 3 cm, and 80 cases (63.5%) were giant CPs (>3 cm). Fifty-one patients (41.5%) had predominantly cystic lesions, 26 patients (20.6%) showed mainly solid lesions, and the remaining 49 patients (38.9%) manifested mixed solid and cystic lesions. Only 21 cases were Puget grade 0, 55 (43.7%) were grade 1, and 50 (39.7%) had lesions involving the hypothalamus (grade 2).

### 3.3. Endocrinological Function

The pre- and postoperative pituitary hormone levels and replacement treatments are shown in Figure 1. Before surgery, 83 patients (65.9%) were diagnosed with functional deficiency of the anterior pituitary, 36 (28.6%) with CH, 69 (54.8%) with GHD, 65 (51.6%) with HPAD, 49 (38.9%) with HPTD, and 56 (44.4%) with GD. Fourteen patients (11.1%) had DI caused by a posterior pituitary hormone deficiency. 

Postoperative follow-up monitoring showed varying degrees of pituitary hormone deficiency in 106 patients (84.1%); 67.5% of them (85 cases) had panhypopituitarism, and all 52 patients with pituitary stalk interruptions exhibited panhypopituitarism (shown in Figure 2). Postoperative single hormone deficiency included GHD (103 cases, 81.7%), HPAD (100 cases, 79.4%), HPTD (90 cases, 71.4%), and GD (81 cases, 64.3%). In all, 118 patients (93.6%) had postoperative transient diabetes insipidus, which lasted for more than 1 year in 29 patients (23%) and required vasopressin to control the amount of urine. Postoperatively, 53.5% (23/43) of patients with normal preoperative pituitary hormones had at least one new deficit; 54.4% (49/90) of patients were newly panhypopituitarism; 59.6% (34/57) of patients were newly GHD; 57.4% (35/61) of patients were newly HPAD; 53.2% (41/77) of patients were newly HPTD; 35.7% (25/60) of patients were newly GD; and 92.9% (104/112) of patients were newly DI. Only 18 (14.3%) patients with normal preoperative pituitary hormones retained normal after surgery (shown in Figure 3). 

Only two patients with anterior pituitary hormone deficiency recovered after surgery in our cohort. One of them had a lower level of preoperative ACTH. Another patient had panhypopituitarism before surgery, and preoperative MR images revealed an intrasellar CP with a stroke. The pituitary stalk was not invaded by the tumor and was completely retained, and the endocrine function of this patient returned to normal after transsphenoidal resection (shown in Figure 4).

Replacement treatment was implemented in 65.6% of patients with HPTD, 46% with cortisol deficiency, and 23.5% and 20.4% with GD and GHD, respectively; only patients with DI received replacement treatment.

### 3.4. Surgical Strategies and Outcomes

Craniotomy was performed in 104 patients (82.5%), and resection through transsphenoidal endoscopy or microscopy was performed in 22 patients (17.5%). The pituitary stalk was completely retained in 74 patients (58.7%). Postoperative imaging confirmed that 80 patients underwent total resection (63.5%). According to postoperative pathological analysis, 73.8% of the patients were adamantinomatous, and 26.2% were papillary. Thirty-one patients (24.9%) relapsed during the postoperative follow-up period.

### 3.5. The Difference in Clinical Presentation and Results between Children and Adults

Compared to the children-onset (CO) group, the adult-onset (AO) group more frequently presented with the symptom of visual impairment (52.6% vs. 24.1%), while the ratio of tumor size more than 3 cm was higher in the CO group (55.2% vs. 30.9%), suprasellar extension reaching the ventricle was also more frequently present in the CO group, pre- and postoperative PHD were significantly more prevalent in the CO group, especially CH (86.2% vs. 61.9%), as well as the postoperative transient DI. The results of operation showed that the AO group more easily got GTR and STR than the CO group (*p* < 0.01).

### 3.6. Risk Factors for Endocrine Dysfunction in Patients

The univariate analysis of preoperative pituitary hormone levels is shown in Table 2. There were differences between children and adults (*p* = 0.008); patients with hormone deficiency-related symptoms (*p* = 0.000) and hypothalamic-associated symptoms (*p* = 0.016) had a higher ratio of hormone deficiency. Imaging data analysis showed that factors influencing preoperative hormone levels included tumor size, location, and degree of hypothalamic involvement. Multiple logistic regression analysis showed that children were more susceptible to preoperative endocrine hormone deficiency (OR = 3.024, *p* = 0.024; 95% CI = 1.21–13.44), and patients with hypothalamic involvement had a high risk of preoperative hormone deficiency (Puget grade 1 vs. grade 0, OR = 4.974, *p* = 0.014; 95% CI = 1.39–17.81) (Puget grade 2 vs. grade 0, OR = 11.452, *p* = 0.000; 95% CI = 2.98–43.97) (Table 3). 

The univariate analysis of the 74 patients with a retained pituitary stalk is shown in Table 4. The results indicate that the factors influencing postoperative hormone deficiency include the position of the lesion, degree of hypothalamus involvement, preoperative hormone condition, and surgical approaches. Multiple logistic regression analysis indicated that patients with suprasellar lesions had a high risk of postoperative hormone deficiency (Suprasellar-extraventricular vs. Intrasellar, OR = 39.427, *p* = 0.002; 95% CI = 3.79–410.22) (Suprasellar-intraventricular vs. Intrasellar, OR = 40.140, *p* = 0.008; 95% CI = 2.67–602.66), and patients with preoperative hormone deficiency had a higher risk of hormone deficiency after surgery (OR = 41.384, *p* = 0.001; 95% CI = 4.50–380.60) (Table 5).

## 4. Discussion

Most patients with CP already have pituitary hormone deficiencies at the time of diagnosis; it has been reported that 40–87% of patients had at least one endocrine hormone deficit, and 17–27% of patients had central DI at the time of diagnosis [13,14]. Postoperatively, endocrine deficiencies of patients with CP are usually aggravated; more than 50% of patients with normal endocrine hormones develop a new deficit of the hypothalamic–pituitary axis after surgery, 80–90% of patients with panhypopituitarism [15,16]. A 10-year follow-up study of patients with CP after surgery showed that the ratios of GH, FSH/LH, ACTH, and TSH deficiency were 88%, 90%, 86%, and 80%, respectively, and DI was about 65% [17]. Consistent with previous reports, in this study, 65.9% of the patients had preoperative anterior pituitary hormone deficiency, and 28.6% had panhypopituitarism. Postoperatively, the percentage of patients with hormone deficiency increased to 84.1%, and 67.5% presented panhypopituitarism. Of the patients in this study, 11.1% had preoperative central DI, while postoperative transient central DI was as high as 93.6%. Although treatment methods and surgical safety are improving daily, patients with CPs still have a high proportion of long-term endocrine defects, which leads to a poor prognosis and is a challenge for CP treatment.

In this study, we analyzed the factors influencing pre- and postoperative hormonal status in 126 patients with CP. Univariate analysis of preoperative hormone status indicated that patients with hypothalamus-related endocrine symptoms had a higher proportion of preoperative pituitary hormone deficiency. Imaging analysis showed that patients with extrasellar lesions, lesions > 3 cm in size, and lesions with hypothalamic involvement (Puget grades 1 and 2) had a higher ratio of preoperative endocrine defects. All patients with preoperative endocrine-related symptoms had different degrees of pituitary hormone deficiency, as confirmed by laboratory examinations. Capatina et al. performed a retrospective analysis of 107 CP cases (72 adults, 35 children) and found that children are more likely to have endocrine hormone deficiency at diagnosis than adults (GHD (68.8% vs. 17.1%), DI (28.5% vs. 8.3%)) [18]. Gautier et al. found that 10–18-year-old children were more prone to hormone deficiency by analyzing 171 patients with CP [19]. Consistent with these previous reports, we also revealed that children had a higher risk of endocrine hormone deficiency at the time of diagnosis. Most patients with hypothalamic involvement had hypothalamic–pituitary axis invasion and destruction, especially those with Puget grade 2. In our group, 82% (41/50) of Puget grade 2 patients had preoperative endocrine hormone deficiency; studies have also confirmed that hypothalamic involvement is associated with panhypopituitarism [20]. 

Univariate analysis of patients’ postoperative hormonal status in our group demonstrated that location, size, degree of hypothalamic involvement, surgical methods, and preoperative hormonal status were the main influencing factors. Multivariate regression analysis indicated that preoperative hormonal status and lesion location were the main factors influencing postoperative hormonal status. It has been widely reported that endocrine defects of CPs are extremely rare to recover, which is distinct from those of pituitary tumors [17,21,22,23]. Different to pituitary tumors, CP has large lesions at diagnosis, which severely destroy the hypothalamus–pituitary system and lead to permanent endocrine disorders. Surgical resection of the tumor cannot recover the endocrine function of patients. Only two patients with preoperative hormone deficiency recovered after surgery in our cohort; the pituitary stalk was not invaded by the tumor and was completely retained. We believe that the preoperative hormone deficiency was transient and was due to acute pituitary ischemia caused by acute compression. After surgery, the compression was relieved, and the pituitary function recovered. The preoperative hormone levels of the other 18 patients with normal postoperative hormone levels were also normal. These results indicate that preoperative hormone status is one of the main factors influencing postoperative hormone levels, and that preoperative hormone deficiency in patients with CP rarely leads to postoperative recovery. With the development of endoscopic technology, the endonasal endoscopic approach (EEA) has been widely used in CP surgery. The EEA has no limitations of visualization, which can be extended to the transplanum and transtuberculum approaches that provide direct midline access to sellar, supra-sellar, retrochiasmatic, and third ventricular tumors [24]. In recent years, the EEA has been used for not only purely sellar CP but also suprasellar CP, even with intra-ventricular extension [25]. The detailed surgical view of the surrounding nerve and vascular structures provided by endoscopy makes it easy for total resection of the tumor. Dho reported a group of patients undergoing EEA for craniopharyngioma; the rate of GTR was up to 91.1% (62/68) [26]. EEA also has advantages in endocrine function preservation, as it has been reported that TCA results in increased new hypopituitarism versus EEA (57.1% vs. 28.9%) [27]. Based on an analysis of 112 cases of surgically treated CPs, Mortini reported that patients who underwent transsphenoidal surgery had a lower risk of de novo endocrine hormone deficiencies [4]. In addition, studies have shown that EEA preserves hypothalamic function regardless of preoperative hypothalamic invasion [28]. The univariate analysis in this group also confirmed that, compared to craniotomy, the TSA had a low ratio of hormone deficiency. Multivariate analysis revealed that intrasellar lesions were associated with a lower risk of hormone deficiency. We usually choose the TSA for intrasellar lesions because it is less involved in the hypothalamic–pituitary axis (mostly Puget Grade 0), and the pituitary stalk is easier to retain during surgery, which has less effect on endocrine function. In the future, EEA should be considered the most important choice for CP to achieve better outcomes and fewer pituitary hormonal disorders [29].

Achieving tumor control and function preservation simultaneously is a challenge for CP treatment. Residual lesions are the main risk factors for CP recurrence. Some scholars believe that the pituitary stalk should be resected to avoid residual lesions and to prevent recurrence accompanied by hormone replacement therapy [4,23,30]. However, other researchers believe that the pituitary stalk should be retained as much as possible to preserve endocrine function, especially when the tumor does not invade the pituitary stalk [31]. Li et al. reported that pituitary stalk preservation reduced endocrine changes in 420 surgically treated CPs without increasing tumor recurrence [32]. In this study, the pituitary stalk was sacrificed during surgery in 52 patients (41.3%), all of whom experienced panhypopituitarism after the operation. However, whether the pituitary stalk should be retained during CP surgery remains controversial. According to previous reports and this study, surgery cannot reverse existing hormone deficiencies in patients with CP. Therefore, we believe that preoperative panhypopituitarism can serve as an indication for pituitary stalk scarification, but for patients with normal preoperative hormones, intrasellar lesions, and without hypothalamic–pituitary axis involvement, the pituitary stalk should be retained as much as possible to preserve endocrine function.

Postoperative hypothalamic damage is a common complication for CP, mainly related to increased body weight, altered circadian rhythms, behavioral changes, sleep irregularities, and so on; the rate is up to 65–80% in CP patients after GTR [33]. The Puget Classification was chosen to help surgeons classify tumors and avoid hypothalamic injury. Patients with no hypothalamic involvement (Grade 0) should attempt GTR, while those with clear hypothalamic invasion (Grade 2) should undergo subtotal re-section, and combined with adjuvant radiation therapy [27]. Many studies have also revealed that the risk of recurrence after GTR vs. STR + radiation therapy did not reach significance [34]. However, endocrinopathy is the most common late toxicity for radiation therapy. Rachel reported a group of pediatric patients treated with surgery and proton therapy and found that the surgery increased the rates of endocrine deficiency from 40% to 87%, complete hormone deficiency increased from 30% to 36%, while, after a median follow-up of 4.8 years from the initiation of RT treatment, up to 94% of patients had some form of endocrinopathy, and 56% had panhypopituitarism [35]. Another study also revealed that radiation therapy is effective in reducing the recurrence rates of STR-treated CPs but increases the rates of endocrine dysfunction [14]. In this study, most patients had endocrinopathies, and RT was less frequently recommended, but the potential adverse effects of RT on endocrine cannot be neglected.

Hormone deficiency in patients with CP is mostly life-long, and CP is known as a life-changing tumor [36]. Long-term hormone deficiency severely influences the development and mental health of patients, particularly children. In this study, we also revealed that endocrinopathy was more common in children, especially panhypopituitarism. Childhood and adolescence are important stages in the physical development and maturation of secondary sexual characteristics, which require a variety of hormones to play a synergistic role in achieving natural physical development. Normal pituitary hormone replacement is critical to decreasing the morbidity associated with the endocrine and metabolic consequences of CP [37]. Therefore, we must pay more attention to hormone deficiency in children with CP. In addition, studies have shown that GH deficiency could increase the risk of cardiovascular disease and that sex hormone deficiency could increase the risk of osteoporosis [38,39]. Although there is still controversy about replacement therapy for endocrine hormone deficiency after brain tumor surgery, at present, replacement therapy remains the treatment option for endocrine deficiency [40]. Supplementation with hormones, including GH, thyroid hormones, and sex hormones, can improve insulin resistance, control body weight, and improve quality of life without increasing the risk of tumor recurrence [13,17,41,42]. The long-term follow-up of the patients in this study indicated that the proportion of patients who accomplished standardized and continuous endocrine replacement therapy was low; the replacement therapy of sex hormones and GH was only approximately 20%. 

It remains a challenging tumor for CP to manage. Hypothalamic–pituitary axis related to tumor control, body’s growth, development, metabolism, reproduction, and psychology. With increasing recognition of the impacts of both tumors and the consequences of their management, we should focus on treatments that not only improve survival but also reduce the adverse impact of hypothalamic morbidity. Regardless of treatment choice, life-long multidisciplinary care is essential to surveil for recurrence, as well as to manage and offer rehabilitation to those who are left with endocrine, metabolic, visual, neurocognitive, and psychosocial sequelae [40,43]. Replacement therapy for long-term endocrine hormone deficiency in patients with CP should be guided by physicians from neurosurgery, endocrinology, pediatrics, radiology, neurooncology, and psychology. They consisted of a specialized multidisciplinary team and made individualized replacement—treatment strategies for each patient, as this can improve the quality of life and long-term prognosis of CP patients.

## 5. Conclusions

Children and patients with hypothalamus-involved tumors are more likely to have hormone deficiency at diagnosis. Surgical resection cannot correct preoperative hormone deficiency, and preoperative hormone status is the main factor influencing postoperative endocrine function. Preservation of the pituitary stalk helps prevent pituitary hormone deficiency, but in patients with preoperative panhypopituitarism, the pituitary stalk can be sacrificed to achieve total resection. Simultaneously, multidisciplinary cooperation should be strengthened to guide replacement therapy for patients with CP and long-term postoperative endocrine hormone deficiency. A clinical algorithm for managing hormone deficiency in patients with craniopharyngioma is shown in Figure 5.

## Figures and Tables

**Figure 1 cancers-15-00340-f001:**
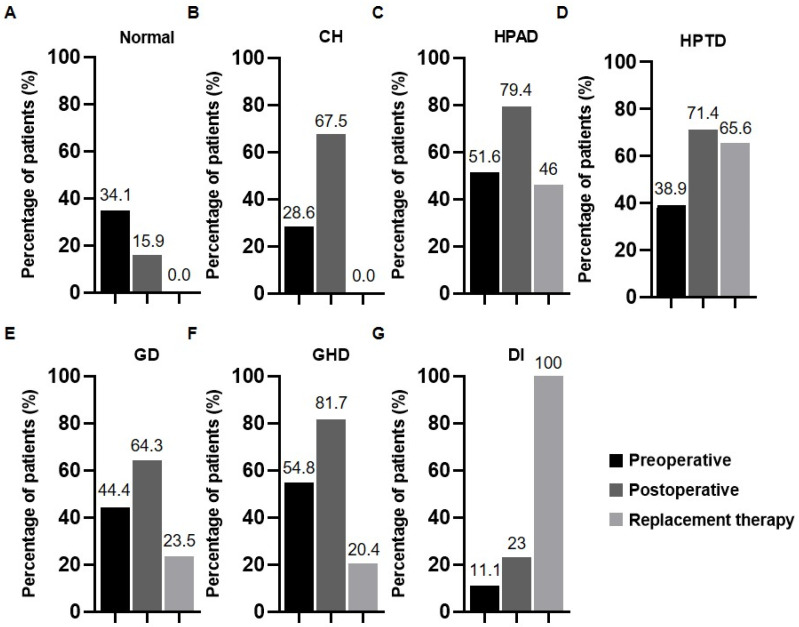
Endocrine deficiencies of the hypothalamic–pituitary axis and replacement therapies in patients with craniopharyngioma. (**A**) The percentage of patients with normal pre- and postoperative pituitary hormone. (**B**) The percentage of patients with pre- and postoperative complete hypopituitarism. (**C**) The percentage of patients with pre- and postoperative hypothalamic–pituitary–adrenal axis deficiency, as well as the percentage of patients with replacement treatment. (**D**) The percentage of patients with pre- and postoperative hypothalamic–pituitary–thyroid axis deficiency, as well as the percentage of patients with replacement treatment. (**E**) The percentage of patients with pre- and postoperative gonadotrophin deficiency axis deficiency, as well as the percentage of patients with replacement treatment. (**F**) The percentage of patients with pre- and postoperative growth hormone deficiency, as well as the percentage of patients with replacement treatment. (**G**) The percentage of patients with pre- and postoperative diabetes insipidus, as well as the percentage of patients with replacement treatment. CH: Complete hypopituitarism, HPAD: Hypothalamic–pituitary–adrenal axis deficiency, HPTD: Hypothalamic–pituitary–thyroid axis deficiency, GD: Gonadotrophin deficiency, GHD: Growth hormone deficiency, DI: Diabetes insipidus.

**Figure 2 cancers-15-00340-f002:**
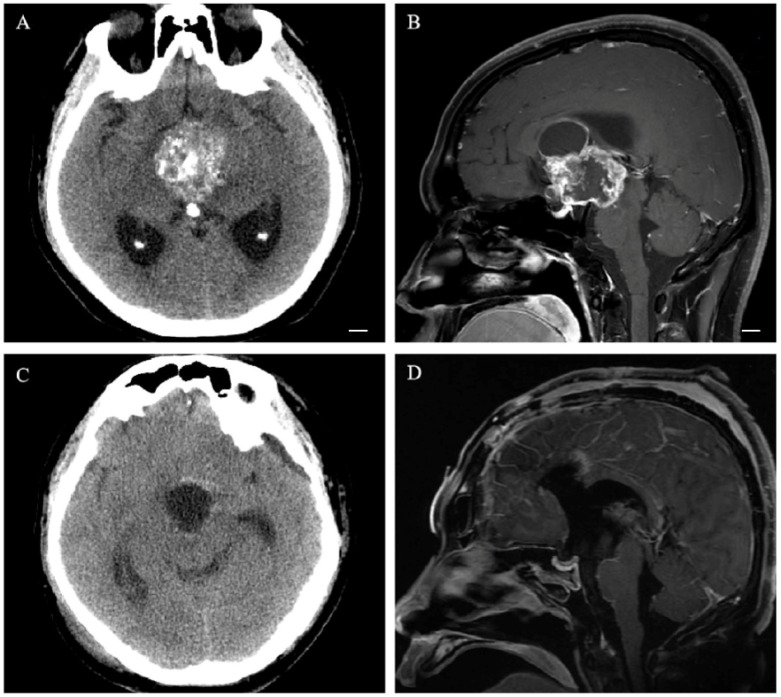
A 35-year-old male with apathy for 2 years accompanied by vision loss for 1 month presented with preoperative panhypopituitarism. CT MRI showed a solid-cystic space-occupying lesion with calcification in suprasellar, the third ventricle and the lateral ventricle (**A**,**B**). The hypothalamus was involved in the tumor (Puget grade 2). The pituitary stalk was invaded and encased by the tumor and was removed during surgery. The tumor and calcification were completely resected (**C**,**D**), and the patient developed panhypopituitarism and diabetes insipidus after surgery. Scale bar: 1 cm.

**Figure 3 cancers-15-00340-f003:**
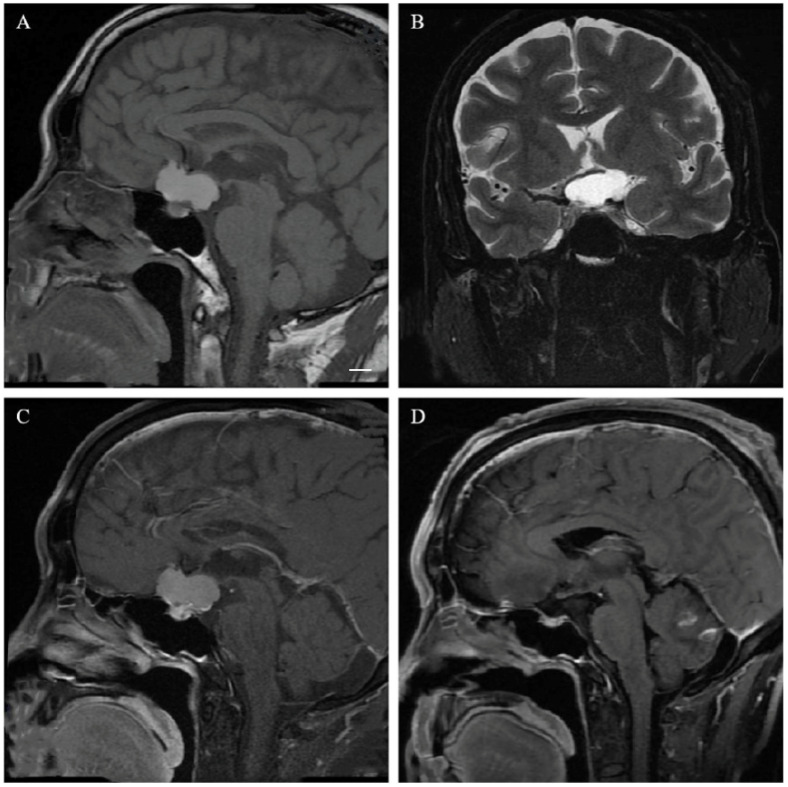
A 61-year-old male, vision loss for 3 months with normal endocrine hormone levels. An MRI showed a suprasellar cystic space-occupying lesion. Slightly hyperintense signal on preoperative T1 and T2 weighted imaging (**A**,**B**), slightly enhanced on T1-weighted contrast-enhanced MR image (**C**). The hypothalamus was involved in the tumor, but the hypothalamus was still identifiable (Puget grade 1). The pituitary stalk was identified and preserved during surgery, and postoperative MRI confirmed total resection (**D**). Postoperative endocrine hormone levels were normal, with the absence of diabetes insipidus. Scale bar: 1 cm.

**Figure 4 cancers-15-00340-f004:**
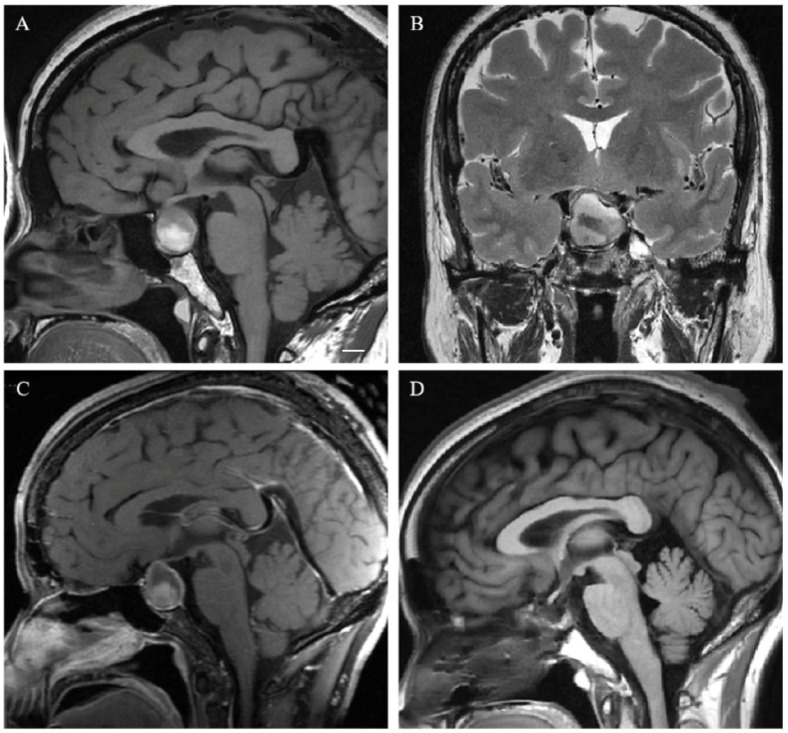
A 63-year-old female with sudden vision loss for three days with preoperative panhypopituitarism. Intrasellar mixed signs are shown on preoperative T1 and T2 weighted imaging (**A**,**B**), and heterogeneous enhancement on T1-weighted contrast-enhanced MR image (**C**). The tumor without hypothalamic involvement (Puget grade 0) was totally resected through transsphenoidal surgery, and the pituitary stalk was completely retained (**D**). Visual loss and hypopituitarism were corrected after surgery. Scale bar: 1 cm.

**Figure 5 cancers-15-00340-f005:**
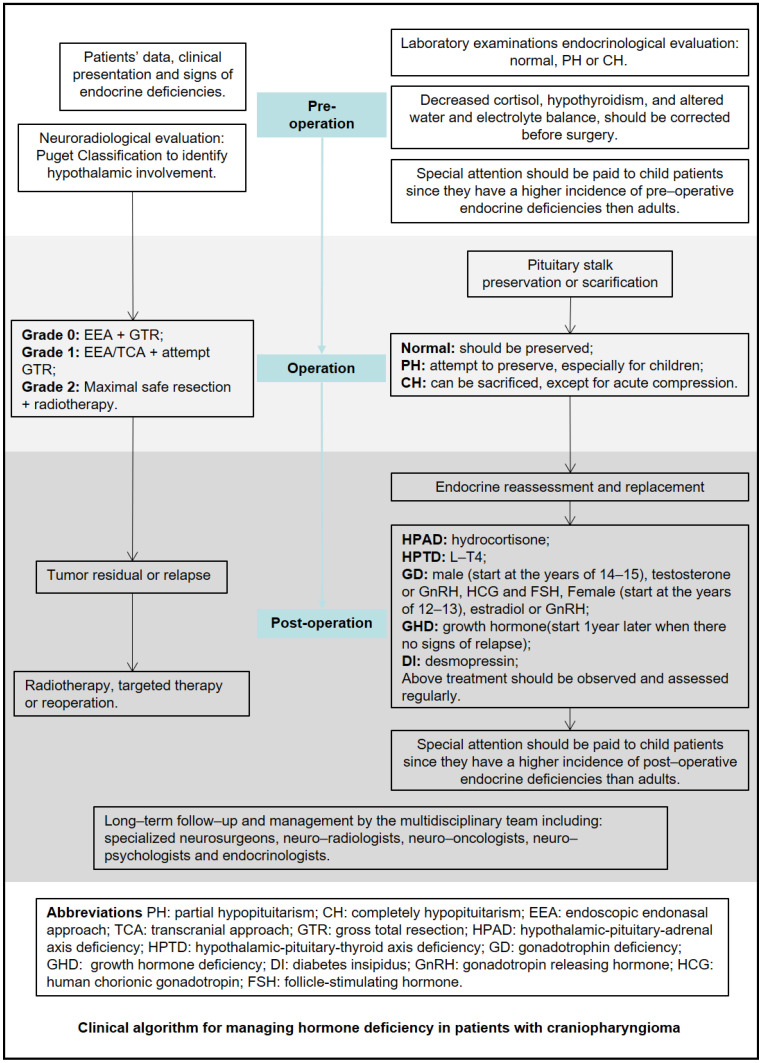
Clinical algorithm for managing hormone deficiency in patients with craniopharyngioma.

**Table 1 cancers-15-00340-t001:** Demographics and clinical characteristics of the patients.

Characteristics	Total (n = 126), (%)	Children (n = 29)	Adults (n = 97)	χ^2^/τ	*p* Value
Age (years)					
Mean ± SD	35.4 ± 19.3	13.4 ± 3.4	41.9 ± 14.0		
Sex					
Male	47 (37.3%)	21 (72.4%)	58 (59.8%)	1.57	0.21
Female	79 (62.7%)	8 (27.6%)	39 (40.2%)		
Clinical manifestation					
Intracranial hypertension	64 (50.8%)	13 (44.8%)	51 (52.6%)	0.537	0.464
visual change	58 (46%)	7 (24.1%)	51 (52.6%)	7.611	0.006 ***
Endocrine symptoms	19 (15.1%)	3 (10.3%)	16 (16.5%)	0.654	0.56
Hypothalamic symptom	11 (8.7%)	2 (6.9%)	9 (9.3%)	0.158	1
Others	15 (11.9%)	7 (24.1%)	8 (8.2%)	4.686	0.030 **
Operation frequency					
Initial operation	109 (86.5%)	26 (89.7%)	83 (85.6%)	0.317	0.76
Reoperation	17 (13.5%)	3 (10.3%)	14 (14.4%)		
Size of lesion					
≤3 cm	46 (36.5%)	16 (55.2%)	30 (30.9%)	5.012	0.025 **
>3 cm	80 (63.5%)	13 (44.8%)	67 (69.1%)		
Tumor texture					
solid	51 (40.5%)	11 (37.9%)	40 (41.2%)	2.416	0.189
cystic-solid	49 (38.9%)	13 (44.9%)	36 (37.1%)	0.553	0.457
cystic	26 (20.6%)	5 (17.2%)	21 (21.7%)	0.294	0.588
Puget hypothalamic involvement					
Grade 0	21 (16.6%)	2 (6.9%)	19 (19.6%)	2.568	0.156
Grade 1	55 (43.7%)	14 (48.3%)	41 (42.3%)	0.326	0.569
Grade 2	50 (39.7%)	13 (44.8%)	37 (38.1%)	0.413	0.52
Location of tumor					
Intrasellar	16 (12.7%)	1 (3.4%)	15 (15.5%)	2.884	0.116
Suprasellar-extraventricular	64 (50.8%)	12 (41.4%)	52 (53.6%)	1.34	0.247
Suprasellar-intraventricular	46 (36.5%)	16 (55.2%)	30 (30.9%)	5.499	0.019 **
Anterior pituitary hormone deficiencies					
Pre-operation	83 (65.9%)				
HPAD	65 (51.6%)	21 (72.4%)	44 (45.4%)	6.75	0.009 ***
HPTD	49 (38.9%)	16 (55.2%)	33 (34.0%)	4.12	0.042
GHD	69 (54.8%)	20 (69.0%)	49 (50.5%)	3.144	0.076
GD(Male > 14 years, Female > 13 years)	56 (44.4%)	19 (65.5%)	37 (38.1%)	6.572	0.010 **
CH	36 (28.6%)	13 (44.8%)	23 (23.7%)	4.613	0.032 **
Post-operation	106 (84.1%)				
HPAD	100 (79.4%)	28 (96.6%)	72 (74.2%)	6.74	0.008 ***
HPTD	90 (71.4%)	24 (82.8%)	66 (68.0%)	2.369	0.11
GHD	103 (81.7%)	28 (96.6%)	75 (77.3%)	5.545	0.063
GD	81(64.3%)	25 (86.2%)	56 (57.7%)	7.822	0.007
CH	85 (67.5%)	25 (86.2%)	60 (61.9%)	5.983	0.014 **
Diabetes insipidus					
Pre-operation	14 (11.1%)	2 (6.9%)	12 (12.4%)	0.672	0.52
Post-operation					
Transient	118 (93.6%)	29 (100%)	89 (91.8%)	4.671	0.038 **
Permanent	29 (23%)	8 (27.6%)	21 (21.6%)	0.444	0.505
Operation method					
Transsphenoidal	22 (17.5%)	4 (13.8%)	18 (18.6%)	0.349	0.781
Craniotomy	104 (82.5%)	25 (86.2%)	79 (81.4%)		
Preservation of pituitary stalk					
Yes	74 (58.7%)	16 (55.2%)	58 (59.8%)	0.197	0.658
No	52 (41.3%)	13 (44.8%)	39 (40.2%)		
Degree of resection					
Total	80 (63.5%)	11 (37.9%)	69 (71.2%)	9.88	0.002 ***
Subtotal	29 (36.5%)	15 (51.7%)	14 (14.4%)	16.181	0.000 ***
Partial	17 (13.5%)	3 (10.4%)	14 (14.4%)	0.048	1
Pathological subtype					
Adamantinomatous	93 (73.8%)	26 (89.7%)	67 (69.1%)	4.854	0.030 **
Papillary	33 (26.2%)	3 (10.3%)	30 (30.9%)		
Hormone alternative therapy					
Yes	80 (63.5%)	24 (82.8%)	56 (57.7%)	6.033	0.014 **
No	46 (36.5%)	5 (17.2%)	41 (42.3%)		
follow-up period (Mean ± SD, month)	54.1 ± 33	52.7 ± 34.8	54.5 ± 32.7	−0.248	0.439
Recurrence					
Yes	31 (24.6%)	8 (27.6%)	23 (23.7%)	0.181	0.674
No	95 (75.4%)	21 (72.4%)	74 (76.3%)		

** *p* < 0.05, *** *p* < 0.01.

**Table 2 cancers-15-00340-t002:** Univariate analysis of factors related to preoperative hypopituitarism in craniopharyngioma patients and age stratification (child and adult at diagnosis).

Variable	Pre-Operation (%)		χ^2^	*p* Value	With Age Stratification	
	With Hypopituitarism	Without Hypopituitarism			HR 95 % CI	*p* Value
Sex						
Male	56 (70.9%)	23 (29.1%)				
Female	27 (57.4%)	20 (42.6%)	2.368	0.124		
Age (years)						
Children (<18)	25 (86.2%)	4 (13.8%)	6.928	0.008 ***		
Adults (≥18)	58 (59.8%)	39 (40.2%)				
Clinical manifestation						
Intracranial hypertension	40 (62.5%)	24 (37.5%)	0.658	0.417		
visual change	34 (58.6%)	24 (41.4%)	2.514	0.113		
Endocrine symptoms	19 (100%)	0 (0%)	11.591	0.000 ***	1.3 (1.1–1.5)	0.001 ***
Hypothalamic symptom	11 (100%)	0 (0%)	6.244	0.016 **	1.2 (1.1–1.3)	0.013 **
Others	12 (80%)	3 (20%)	1.512	0.261		
Location of tumor						
Intrasellar	4 (25%)	12 (75%)	13.916	0.001 ***		
Suprasellar-extraventricular	42 (65.6%)	22 (34.4%)			1.0 (0.5–2.0)	0.953
Suprasellar-intraventricular	37 (92.5%)	9 (7.5%)			3.0 (1.3–7.1)	0.009 ***
Size of lesion						
≤3 cm	24 (%)	22 (%)	6.048	0.014 **	0.4 (0.2–0.8)	0.014 **
>3 cm	59 (%)	21 (%)				
Tumor texture						
solid	36 (70.6%)	15 (29.4%)	0.847	0.357		
cystic-solid	32 (65.3%)	17 (34.7%)				
cystic	15 (57.7%)	11 (42.3%)				
Puget hypothalamic involvement						
Grade 0	4 (19.5%)	17 (81.0%)	22.04	0.000 ***		
Grade 1	38 (69.1%)	17 (30.9%)			1.3 (1.1–2.7)	0.031 **
Grade 2	41 (82%)	9 (18%)			3.7 (1.6–8.6)	0.002 ***
Pathological subtype						
Adamantinomatous	63 (67.7%)	30 (32.3%)	0.552	0.458		
Papillary	20 (60.6%)	13 (39.4%)				

** *p* < 0.05, *** *p* < 0.01.

**Table 3 cancers-15-00340-t003:** Logistic regression analysis of factors related to preoperative hypopituitarism in craniopharyngioma patients.

Variable	OR	95% CI	*p* Value
Age (years)			
Children (<18)	3.024	1.21–13.44	0.024
Clinical manifestation			
Endocrine symptoms			NS
Hypothalamic symptom			NA
Location of tumor			NA
Size of lesion			NA
Puget hypothalamic involvement			
Grade 1 vs. Grade 0	4.974	1.39–17.81	0.014
Grade 2 vs. Grade 0	11.452	2.98–43.97	0

NS = not significant, NA = not applicable.

**Table 4 cancers-15-00340-t004:** Univariate analysis of factors related to postoperative hypopituitarism in 74 craniopharyngioma patients with reserved pituitary stalk, age stratification (child and adult at diagnosis).

Variable	Post-Operation (%)		χ^2^	*p* Value	With Age Stratification	
	With Hypopituitarism	Without Hypopituitarism			HR 95 % CI	*p* Value
Sex						
Male	34 (81%)	8 (19%)	3.136	0.077 *	0.4 (0.1–1.1)	0.077
Female	20 (62.5%)	12 (37.5%)				
Age (years)						
Children (<18)	15 (93.8%)	1 (6.2%)	4.468	0.054 *		
Adults (≥18)	39 (67.2%)	19 (32.8%)				
Clinical manifestation						
Intracranial hypertension	28 (73.7%)	10 (26.3%)	0.02	0.887		
visual change	27 (65.9%)	14 (34.1%)	2.363	0.124		
Endocrine symptoms	10 (100%)	0 (0%)	4.282	0.053 *	1.2 (1.1–1.4)	0.040 **
Hypothalamic symptom	5 (100%)	0 (0%)	1.986	0.159		
Others	5 (83.3%)	1 (16.7%)	0.355	1		
Location of tumor						
Intrasellar	3 (18.8%)	13 (81.2%)	26.781	0.000 ***		
Suprasellar-extraventricular	36 (87.8%)	5 (12.2%)			6.0 (1.9–19.1)	0.001 ***
Suprasellar-intraventricular	15 (88.2%)	2 (11.8%)			3.5 (1.7–16.8)	0.003 ***
Size of lesion						
≤3 cm	23 (62.2%)	14 (37.8%)	4.385	0.036 **	0.3 (0.1–0.9)	0.036 **
>3 cm	31 (83.8%)	6 (16.2%)				
Tumor texture						
solid	27 (75%)	9 (25%)	1.585	0.453		
cystic-solid	20 (76.9%)	6 (23.1%)				
cystic	7 (58.3%)	5 (41.7%)				
Puget hypothalamic involvement						
Grade 0	6 (28.6%)	15 (71.4%)	29.359	0.000 ***		
Grade 1	24 (80.6%)	3 (19.4%)			4.5 (1.2–17.3)	0.020 **
Grade 2	24 (92.3%)	2 (7.7%)			7.2 (1.5–34.1)	0.006 ***
Preoperative hormonal status						
With hypopituitarism	41 (95.3%)	2 (4.7%)	26.058	0.000 ***	28.4 (5.8–138.9)	0.000 ***
Without hypopituitarism	13 (41.9%)	18 (58.1%)				
Operation frequency						
Initial operation	6 (85.7%)	1 (14.3%)	0.636	0.666		
Reoperation	48 (71.6%)	19 (28.4%)				
Operation method						
Transsphenoidal	8 (36.4%)	14 (63.6%)	21.275	0.000 ***	0.1 (0.0–0.3)	0.000 ***
Craniotomy	46 (88.5%)	6 (11.5%)				
Degree of resection						
Total	35 (71.4%)	14 (28.6%)	0.675	0.175		
Subtotal	14 (87.5%)	2 (12.5%)	2.184	0.207		
Partial	5 (55.6%)	4 (44.4%)	1.576	0.241		
Pathological subtype						
Adamantinomatous	14 (66.7%)	7 (33.3%)	0.591	0.442		
Papillary	40 (75.5%)	13 (24.5%)				

* *p* < 0.1, ** *p* < 0.05, *** *p* < 0.01.

**Table 5 cancers-15-00340-t005:** Logistic regression analysis of factors related to postoperative hypopituitarism in craniopharyngioma patients.

Variable	OR	95% CI	*p* Value
Sex			NA
Age (years)			NA
Clinical manifestation			
Endocrine symptoms			NA
Hypothalamic symptom			NA
Location of tumor			
Suprasellar-extraventricular vs. Intrasellar	39.427	3.79–410.22	0.002
Suprasellar-intraventricular vs. Intrasellar	40.14	2.67–602.66	0.008
Size of lesion			NS
Puget hypothalamic involvement			NS
Preoperative hormonal status			
With hypopituitarism	41.384	4.50–380.60	0.001
Operation method			NS

NS = not significant, NA = not applicable.

## Data Availability

Data are available on request.

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
