# Peer review of "Risk Factor and Replacement Therapy Analysis of Pre- and Postoperative Endocrine Deficiencies for Craniopharyngioma"

_cancers, 2023, doi:10.3390/cancers15020340_

Round 1

Reviewer 1 Report

We carefully reviewed the present manuscript submitted for publication on Cancers. The authors present an interesting article on the risk factor for pre e post operative endocrine deficiencies in craniopharyngioma. Their main assumption is that pre-operative endocrine dysfunction does not recovery after surgery,  therefore pituitary stalk should be sacrificed, favoring tumor total removal.  This issue is commonly shared in CP treatment strategy, although hormonal replacement therapy brings its severe drawbacks.  The manuscript doesn’t add any new insight to the pertinent literature, however the well done statistical analysis reinforces this attitude.  Please accept the following criticisms:

1.      Text and tables do not always match (please see pre-operative status); also, intrasellar tumor is sometimes considered as a major risk factor for pre-op deficiency and other time it is not. 

2.     References about the endoscopic endonasal approach should involve more relevant publication on the topic.

3.     Considerations should be stratified between the adult and pediatric population, in regards to the higher rate of endocrine deficiency at presentation in children and the different short- and long-term impact on children and adults.

4.     The rate of gross total tumor removal in the presence of a preserved pituitary function is much lower as compared to the group of patients with pre-op endocrine deficiency.  What was the rate of recurrence in these cases? How did recurrence impact on the natural history of the disease?  Five years follow-up can be adequate to answer to this question.

Author Response

Comments: We carefully reviewed the present manuscript submitted for publication on Cancers. The authors present an interesting article on the risk factor for pre and post-operative endocrine deficiencies in craniopharyngioma. Their main assumption is that pre-operative endocrine dysfunction does not recovery after surgery, therefore pituitary stalk should be sacrificed, favoring tumor total removal.  This issue is commonly shared in CP treatment strategy, although hormonal replacement therapy brings its severe drawbacks.  The manuscript doesn’t add any new insight to the pertinent literature, however the well-done statistical analysis reinforces this attitude.  Please accept the following criticisms:

Response to Reviewer 1: We thank for your careful reading, helpful comments, and constructive suggestions, which significantly improved the presentation and quality of our manuscript. We summarize our responses to each comment below. We hope our responses have well addressed all of your concerns.

Q1: Text and tables do not always match (please see pre-operative status); also, intrasellar tumor is sometimes considered as a major risk factor for pre-op deficiency and other time it is not. 

Response to Q1: Thanks for your helpful comments. We apologize for our error in writing and the typos have been corrected (Line 303 in the revised manuscript). As you mentioned here, intrasellar tumor is sometimes considered as a major risk factor for pre-op deficiency and other time it is not. In this study, we found that patients with intrasellar craniopharyngioma have a higher percentage of normal preoperative hormone levels, which is possibly related to less disruption of the hypothalamic-pituitary axis by tumors.

Q2: References about the endoscopic endonasal approach should involve more relevant publication on the topic.

Response to Q2: Thanks for your constructive suggestions. The endoscopic endonasal approach for craniopharyngioma treatments has been mature relatively and the indications for its application are gradually expanded. Based on your suggestions, we have enrolled several relevant publications in the discussion part (Line 335-356 in the revised manuscript).

Q3: Considerations should be stratified between the adult and pediatric population, in regards to the higher rate of endocrine deficiency at presentation in children and the different short- and long-term impact on children and adults.

Response to Q3: Thanks for your helpful comments. The impacts of endocrine hormone deficiency are more prominent in children. In the revised manuscript, patients were divided into the children-onset (CO) and the adult-onset (AO) groups and we analyzed the clinical information and prognosis of patients in each group. To exclude the effect of the confounding factor of age, a stratified chi-square test using the Cochran-Mantel-Haenszel method was performed to analyze factors that are significant in univariate analysis (see part 3.4 (line 234) and Tables 2 and 4 in the revised manuscript)

Q4: The rate of gross total tumor removal in the presence of a preserved pituitary function is much lower as compared to the group of patients with pre-op endocrine deficiency.  What was the rate of recurrence in these cases? How did recurrence impact on the natural history of the disease?  Five years follow-up can be adequate to answer to this question.

Response to Q4: Thanks for your helpful comments. As you mentioned here, the five-year recurrence rate in these cases with a preserved pituitary function was nearly 10% higher than patients with pre-op endocrine deficiency (34.2% vs 25%). The majority of patients with recurrence opted for reoperation.

Reviewer 2 Report

The present study about the pre and postoperative HD in patients managed for CP is well written, conducted and documented; the methods and results sound although finally it does not add neither new knowledges nor new management strategies; In my opinion the authors could eventually greatly improve the paper by:

1)    Creating /developing a clinical algorithm of CP management to help/guide clinicians to decide the best management strategies in their routine practice.

2)    The authors very briefly state the risks of surgery and radiotherapy in managing CP but they do not analyze the results of surgery versus radio or proton-therapy neither in their series nor in the literature (please discuss, add details, explain, argument).

3)    The authors briefly state about the two (children and adults) disease peak of incidence, but they do not present/analyze all clinical differences and peculiarities related to each population discussing also any other optional management strategies and of course the latter could be further introduced in the clinical treatment algorithm (please add, discuss)

4)    The authors do not analyze the different CP growing profile/pattern in children and adults and its own implication on hormonal function dealing to a different/tailored strategy (please discuss, add details).

5)    When the authors discuss about pituitary stalk retention, I would instead discuss about PS preservation/respect/sparing the word retention could be confusing (please change).

6)    Finally CP is a very complex and challenging pathology and the authors never discuss its multidisciplinary management that of course represents the key of success (please discuss, add details).

Author Response

Comments: The present study about the pre and postoperative HD in patients managed for CP is well written, conducted and documented; the methods and results sound although finally it does not add neither new knowledges nor new management strategies; In my opinion the authors could eventually greatly improve the paper by:

Response to Reviewer 2: We thank for your careful reading, helpful comments, and constructive suggestions, which significantly improved the presentation and quality of our manuscript. We summarize our responses to each comment below. We hope our responses have well addressed all of your concerns.

Q1: Creating /developing a clinical algorithm of CP management to help/guide clinicians to decide the best management strategies in their routine practice.

Response to Q1: Thanks for your helpful suggestions. According to our experience in research and clinical, a simple clinical algorithm has been developed at the end of the revised manuscript (Figure 5).

Q2: The authors very briefly state the risks of surgery and radiotherapy in managing CP but they do not analyze the results of surgery versus radio or proton-therapy neither in their series nor in the literature (please discuss, add details, explain, argument).

Response to Q2: Thanks for your helpful suggestions. Since most patients in this study had pituitary hormone deficiency before radiotherapy, and fewer patients were recommended for radiotherapy, we enrolled several relevant publications in the revised manuscript to discuss the impact of radiotherapy on patients’ endocrine functions. (Line 373-390 in the revised manuscript)

Q3: The authors briefly state about the two (children and adults) disease peak of incidence, but they do not present/analyze all clinical differences and peculiarities related to each population discussing also any other optional management strategies and of course the latter could be further introduced in the clinical treatment algorithm (please add, discuss)

Response to Q3: Thanks for your constructive comments. In the revised manuscript, patients were divided into the children-onset (CO) and the adult-onset (AO) groups and we analyzed the clinical information and prognosis of patients in each group. To exclude the effect of the confounding factor of age, a stratified chi-square test using the Cochran-Mantel-Haenszel method was performed to analyze factors that are significant in univariate analysis (see part 3.4 (line 234) and Tables 2 and 4 in the revised manuscript). We found that children had a higher rate of endocrine deficiency at diagnosis and a higher risk of preoperative and postoperative panhypopituitarism, which indicated that more attention should be paid to endocrine deficiency in children patients with craniopharyngioma. This has also been introduced in the clinical treatment algorithm.

Q4: The authors do not analyze the different CP growing profile/pattern in children and adults and its own implication on hormonal function dealing to a different/tailored strategy (please discuss, add details).

Response to Q4: Thanks for your constructive comments. Pediatric patients are often admitted with larger tumors due to relatively insidious symptoms. The lesions of pediatric patients are more likely to involve the ventricles which resulted in a higher percentage of endocrine deficiency. Furthermore, due to the special period of growth and development, postoperative complications have a greater impact on pediatric patients, which requires better surgical skills and postoperative managements. In the revised manuscript, we added analysis and discussion of the different CP growing patterns in children and adults (Line 393-400 in the revised manuscript).

Q5: When the authors discuss about pituitary stalk retention, I would instead discuss about PS preservation/respect/sparing the word retention could be confusing (please change).

Response to Q5: Thanks for your kind suggestions. We have changed ‘pituitary stalk retention’ to ‘pituitary stalk preservation’ in the revised manuscript.

Q6: Finally, CP is a very complex and challenging pathology and the authors never discuss its multidisciplinary management that of course represents the key of success (please discuss, add details).

Response to Q6: Thanks for your constructive suggestions. As you have mentioned here, craniopharyngioma is a very complex and challenging pathology and we need to shift from focusing on treatments only in the past to simultaneously focusing on treatments and long-term managements. In the revised manuscript, we added discussions on the importance of long-term and multidisciplinary management of craniopharyngioma (Line 411-423 in the revised manuscript).

Round 2

Reviewer 1 Report

Authors have properly answered to the review's criticism and the manuscript has been improved. 

Reviewer 2 Report

The authors answered satisfactorily to the reviewers comments.